# HiRID-ICU-Benchmark — A Comprehensive Machine Learning Benchmark on High-resolution ICU Data

**Hugo Yèche** [1] [*]       **Rita Kuznetsova** [1] [*]       **Marc Zimmermann** [1]

**Matthias Hüser** [1]       **Xinrui Lyu** [1]       **Martin Faltys** [1,2]       **Gunnar Rätsch** [1]

{hyeche,mkuznetsova,marczim,mhueser,xlyu,mfaltys,raetsch}@inf.ethz.ch
[1]Department of Computer Science, ETH Zürich
[2]Department of Intensive Care Medicine, University Hospital, and University of Bern

## Abstract

The recent success of machine learning methods applied to time series collected from Intensive Care Units (ICU) exposes the lack of standardized machine learning benchmarks for developing and comparing such methods. While raw datasets, such as MIMIC-IV or eICU, can be freely accessed on Physionet, the choice of tasks and pre-processing is often chosen ad-hoc for each publication, limiting comparability across publications. In this work, we aim to improve this situation by providing a benchmark covering a large spectrum of ICU-related tasks. Using the HiRID dataset, we define multiple clinically relevant tasks in collaboration with clinicians. In addition, we provide a reproducible end-to-end pipeline to construct both data and labels. Finally, we provide an in-depth analysis of current state-of-the-art sequence modeling methods, highlighting some limitations of deep learning approaches for this type of data. With this benchmark, we hope to give the research community the possibility of a fair comparison of their work.

**Software Repository:** https://github.com/ratschlab/HIRID-ICU-Benchmark/

## 1   Introduction

Severely ill patients require treatment and surveillance in Intensive Care Units (ICU). Critical health conditions are characterized by the presence or risk of developing life-threatening organ dysfunction. During a patient's stay in the ICU, continuous monitoring of organs function parameters enables early recognition of physiological deterioration and rapid commencement of appropriate interventions. Recent research shows the great success of machine learning methods when applied to ICU time series [45, 19]. One of the main goals of previous works was to develop new methods for prediction tasks relevant to clinical decision-making. Exemplary of such tasks are alarm systems that predict different types of organ failure [20, 42].

To develop and evaluate such methods only a small number of large-scale ICU datasets are freely-accessible: The MIMIC-III [24] and IV [22] datasets, AmsterdamUMCdb [1], HiRID [12] and the eICU Collaborative Research Database [35]. However, these datasets are not provided in a pre-processed form directly suitable for machine learning nor do they have well-defined tasks, making it impossible to fairly compare works [23]. While some pre-processed alternatives with well-defined tasks exist [14, 36], they are often lacking in terms of size and diversity of tasks. We provide more

---
[*]Equal contribution

35th Conference on Neural Information Processing Systems (NeurIPS 2021) Track on Datasets and Benchmarks.

details about this in section 2. This leads to situations where works compare methods on their private data [42] or only on limited data and number of tasks. Also the lack of relevant clinical sub-tasks for benchmarking hinders the development of new methods for clinical decision support systems [16]. Finally, as in other fields, in recent datasets such as HiRID [12] the time resolution of data has greatly increased. However, no benchmark on ICU time series using such high-resolution datasets currently exists.

To improve this situation, in this paper we provide an in-depth benchmark based on the HiRID dataset [12, 20][2], which was released on Physionet [12] alongside the publication on the circulatory Early Warning Score (circEWS) [20]. HiRID is a freely accessible critical care dataset containing data recorded at the Department of Intensive Care Medicine, the University Hospital of Bern, Switzerland (Inselspital). The dataset was developed in cooperation with the Swiss Federal Institute of Technology (ETH), Zürich, Switzerland. We define a new benchmark on HiRID composed of various clinically relevant tasks and provide a comprehensive pipeline, which includes all steps from preprocessing to model evaluation. To assess different aspects of the benchmarked machine learning methods, we diversify the tasks around specific challenges of ICU data such as prediction frequency, class imbalance, or organ dependency of the task. To profit from data acquisition advances and allow improvement on longer time series, we use a resampled data resolution of 5 min. HiRID has a higher time resolution than any other published critical care dataset and it motivates us to provide a comprehensive benchmark suite on this dataset. Also, we believe that this dataset will facilitate the construction of new predictive methods for the healthcare field, going beyond ICU time series.

The main contributions of this paper are:

- We developed a comprehensive, end-to-end pipeline for time-series analysis of critical care data based on the recently published HiRID dataset. This pipeline includes the following stages: data preprocessing mode, training mode, and evaluation mode.

- We proposed and implemented a variety of tasks relevant to healthcare workers in the ICU, diversified in terms of type, prediction resolution, and label prevalence. The tasks cover all major organ systems as well as the general patient state. We included both regression and classification (binary and multi-class) tasks.

- By providing a comprehensive benchmark on a set of canonical tasks, we give the research community around predictive modeling on ICU time series the possibility for the clear comparison of their methods.

The paper is organized as follows: in Section 2 we provide an overview of existing ICU datasets and benchmarking papers. We provide details about the HiRID dataset and introduce the tasks defined in collaboration with clinicians in Section 3 and give more details on the tasks in APPENDIX A: DATASET DETAILS. Section 4 illustrates the pipeline design, with more details given in APPENDIX B: HIRID-ICU-PIPELINE DETAILS. Section 5 describes the experiment and ablation study. In Section 6 we discuss the observed results and relate this paper to other benchmarks and related tasks relevant for clinicians.

## 2   Related Work

The main goal of this work is to provide a benchmark on the HiRID dataset for various clinical prediction tasks of interest. We describe here other ICU datasets as well as existing benchmarks for ICU data.

**ICU time-series datasets**   There are several widely-used, freely-accessible datasets consisting of ICU time series. MIMIC-III [24] is the oldest and most widely used ICU dataset. It consists of physiological measurements as well as information about laboratory tests. Physiological measurements are recorded with a maximum resolution of 1 hour. The results of laboratory tests are collected at irregular time intervals. Moreover, there are static features like gender, age, diagnosis, etc. available. The dataset consists of information recorded about 40,000 ICU stays at Beth Israel Deaconess Medical Center (BIDMC), Boston, MA, USA. The median of the patient stay length is 2 days. The eICU Collaborative Research Database [35] is a large multicenter critical care database

---

[2]https://physionet.org/content/hirid/1.1.1/

made available by Philips Healthcare in partnership with the MIT Laboratory for Computational Physiology. It contains data associated with over 200,000 patient stays, but the public version does not reach the granularity of other datasets in terms of time resolution and data elements. The first version of AmsterdamUMCdb [1] was released in November 2019. Its current version from March 2020 contains data related to 23,172 ICU and high dependency unit admissions of adult patients from 2003 - 2016 from Amsterdam University Medical Centers. The data includes clinical observations like vital signs, clinical scores, device data, and lab results.

**Benchmarks on ICU time-series.** Among works using the openly available datasets mentioned above, to the best of our knowledge, only a single standardized benchmark exists, MIMIC-III benchmark by Harutyunyan et al. [16]. In that work four tasks were proposed, two requiring a single prediction per patient stay and two dynamic tasks with more frequent prediction, one per hour. In addition, while not proposing a benchmark, Jarrett et al. [21] developed a standardized pipeline for medical time series, called Clairvoyance. They also provided results on several datasets, including MIMIC. In this spirit, some packages address a specific family of tasks, for example, classification [13] and forecasting [15]. Finally, some public challenges, with curated data, were proposed in the past, e.g. the early prediction of sepsis (Physionet 2019 challenge [36]) or mortality prediction (Physionet 2012 challenge [9]). However, the provided datasets are smaller than HiRID and are built around a single task.

# 3 Benchmark Design

## 3.1 The HiRID Dataset

HiRID [12, 20] is a freely accessible critical care dataset containing data from more than 33,000 patient admissions to the Department of Intensive Care Medicine, the University Hospital of Bern, Switzerland (Inselspital) from January 2008 to June 2016. It was released on Physionet [12] alongside the publication of the circulatory Early Warning Score (circEWS) [20]. It contains de-identified demographic information and a total of 712 routinely collected physiological variables, diagnostic test results, and treatment parameters. HiRID has a higher time resolution than any other published ICU dataset, particularly for bedside monitoring, with most vital signs recorded every 2 minutes, which motivates us to provide a comprehensive benchmark suite on this dataset. Demographic information about the patient cohort are displayed in Appendix Table 1.

Table 1: Definition of prediction tasks contained in the HiRID-ICU benchmark suite

| Task name | Task type | Task description |
|---|---|---|
| ICU mortality | Binary classification, one prediction per stay | Predicted at 24h after admission to the ICU. |
| Patient phenotyping | Multi-class classification, one prediction per stay | Classifying the patient after 24h regarding the admission diagnosis, using the APACHE group II and IV labels[3] |
| Circulatory failure [4] | Binary classification, dynamic prediction throughout stay | Continuous prediction of onset of circulatory failure in the next 12h, given the patient is not in failure now. |
| Respiratory failure[5] | Binary classification, dynamic prediction throughout stay | Continuous prediction of onset of respiratory failure in the next 12h, given the patient is not in failure now. |
| Kidney function | Regression, dynamic prediction throughout stay | Continuous prediction of urine production in the next 2h as an average rate in ml/kg/h. The task is predicted at irregular intervals. |
| Remaining length of stay | Regression, dynamic prediction throughout stay | Continuous prediction of the remaining ICU stay duration. |

## 3.2 Prediction Tasks

Our benchmark suite focuses on clinically relevant prediction tasks with a large diversity in the machine learning task types. From a clinical point of view, the tasks cover most major organ systems as well as the general patient state. The major organ systems include the cardiovascular, kidney, and respiratory systems. For each organ system, we provide a prediction task related to the main organ function. Length of stay, mortality, and patient phenotyping are chosen to assess an overall patient state. From a machine learning point of view, our suite contains regression and classification (binary and multi-class) tasks. We included tasks with different degrees of class imbalance to diversify the spectrum further and enable the comparison of methods on e.g. highly imbalanced tasks. We chose tasks performed online throughout the stay (every 5 minutes) and at fixed time-points of the stay, such as 24h after ICU admission, which capture a more long-term state of the patient. To enhance reproducibility, we include two tasks previously considered in [16], mortality, and remaining length-of-stay prediction. Table 1 contains the full list of task and their detailed descriptions.

## 4 Pipeline Design

Figure 1 shows an overview of the major HiRID-ICU pipeline steps. The pipeline is designed using the *preprocess-train-predict* paradigm. We provide more details about it in APPENDIX B: HIRID-ICU PIPELINE DETAILS and the README section of the software repository[6]. The preprocessed data contains two versions, `common_stage` and `ml_stage`. The former is independent of modeling choices and serves as the starting point for future works with custom pre-processing choices. The latter is a compatible version for our pipeline with our categorical encoding, imputation, and scaling choices.

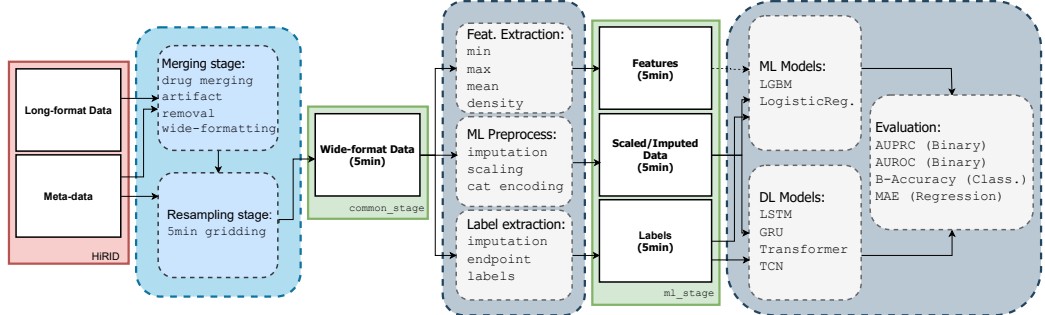

Figure 1: *Detailed Pipeline.* (Red) Raw Long-format Data. (Green) Wide-format data. (Blue) Common data pre-processing. (Grey) Modeling depending stages.

## 4.1 Data Pre-processing

In its public version, HiRID, as any real-world dataset, contains certain artifacts that require pre-processing. As pointed out by [4] for MIMIC-III, individual pre-processing in each work avoids a fair comparison of them. To this effect, we aim to provide a modular and reproducible pipeline. Patient EHRs in HiRID are stored in a long table format where each row of the table is a record containing the measurement value of a specific variable at a specific time for a patient, which cannot be used as a ready input for machine learning models.

**Wide-format Merging** To obtain a more compact format, the first pre-processing step in our pipeline is to transform the long table of patient EHRs into feature matrices, where each column

---

[3]APACHE II and IV [47, 29] are subsequent versions of the major illness severity score used in the ICU. They also introduce a patient grouping according to admission reason. We use an aggregate of these two groupings for this task (see APPENDIX A: DATASET DETAILS)

[4]Circulatory failure is defined as Lactate > 2mmol/l and either mean arterial blood pressure < 65mmHg or administration of any vasoactive drug.

[5]Respiratory failure is defined according to the Berlin definition [2] as a P/F ratio < 300 mmHg.

[6]https://github.com/ratschlab/HIRID-ICU-Benchmark

represents a clinical concept, which we call the wide-format. Such a data format represents an irregularly sampled multivariate time series. At this step, we also remove any physiologically impossible measurement.

**High-Resolution Gridding**   After this merging step, we further compact the dataset by re-sampling it to a 5 minute resolution. Thus, each time step contains the last value measured in the last 5 min if it exists, or left empty otherwise. This gridding strategy is similar to the one used by [20]. We refer to the output of this step as the `common_stage` in Fig.1. Because it is independent of modeling choices, this stage provides a starting point for future approaches using different imputation and scaling choices.

**Processing for Machine Learning**   In the second part of the pipeline, we process the common stage of the data to be compatible with ML models' expected input format. For this, we first use forward-filling imputation for each stay. Then, we apply one-hot encoding for categorical variables and scale the remaining ordinal or continuous variables. We standard scaled all variables with the exception of the time since admission and admission date, which we min-max scaled. By doing scaling globally, we ensure to preserve patients' specificity (e.g.: tachycardia). We refer to the output of this stage as the `ml_stage` as it is dependent on our modeling choices.

## 4.2   Hand-engineered Feature Extraction

In the original paper describing the HiRID dataset [20], the authors showed that boosted tree ensembles such as LGBM [26], when provided with hand-engineered features, outperform state-of-the-art deep learning methods. Based on this observation, we include in our pipeline the possibility to extract such features from the `common_stage` of the data. For our models, we extracted four features for each non-categorical variable over the entire history: *minimum past value*, *maximum past value*, *mean past value* and *density of measurement*, which is the proportion of time points where a value is provided among all possible time points in the history[7]. These features are then included in the `ml_stage`.

## 4.3   Label Construction and Splitting

We construct prediction task labels using the provided measurements and meta-data for both continuous and stay-level tasks. As an intermediate step for label construction, we use a forward imputed version of the data, as in the modeling stage. Concerning the experimental design, we use a random split of patients. The training set contains 70% of the patients and validation and test sets each contain 15%. The temporal splitting strategy as used by Hyland et al. [20] would be more clinically relevant but information about admission time was removed to preserve anonymity when the dataset was originally published. While longer stays exist in the dataset, for computational reasons, we limited labeling to the first 7 days of stays (2016 steps). This cropping affects less than 6% of all stays.

Table 2: Label statistics for each of the tasks, in the training, validation and test sets. As a metric, for binary classification tasks, the positive label prevalence is reported. For multi-class classification tasks, the class prevalence of the minority class is reported. Finally, for regression tasks the median of the label distribution is reported. In parentheses the number of samples is reported. M: Million.

| Task name | Train set | Validation set | Test set |
|---|---|---|---|
| Circ. failure | 4.3 % (n=14.12M) | 4.1 % (n=3.01M) | 4.1 % (n=2.96M) |
| Resp. failure | 38.3 % (n=5.58M) | 37.6 % (n=1.21M) | 37.4 % (n=1.20M) |
| Mortality | 8.7 % (n=10525) | 7.1 % (n=2206) | 8.3 % (n=2231) |
| Phenotyping | 0.2 % (n=10470) | 0.1 % (n=2194) | 0.1 % (n=2217) |
| Kidney function | 1.17 ml/kg/h (n=341424) | 1.12 ml/kg/h (n=71549) | 1.18 ml/kg/h (n=70642) |
| Rem. LOS | 41.04h (n=15.15M) | 41.51h (n=3.22M) | 39.64h (n=3.17M) |

---

[7]This is done on the regularly sampled version of the data

### 4.4 Model Training and Evaluation

The final part of the pipeline contains an end-to-end machine learning suite to train and evaluate our models, depicted on the right hand side of Fig.1.Machine learning (ML) approaches were implemented using `scikit-learn`[34] and `lightgbm`[26], whereas deep learning (DL) approaches were implemented in `pytorch` [33]. All DL models were trained using Adam optimizer [28], with a cross-entropy objective for classification tasks and mean-squared error (MSE) for regression tasks. For classification we provide the possibility to balance loss weights according to class prevalence as in [27].

For the evaluation of models, we use a range of metrics relevant to each task. For classification tasks, we considered AUROC[8] and AUPRC[9] metrics in the binary case, and balanced accuracy (B-Accuracy) [6] in the multi-class one. For regression tasks, we used mean absolute error (MAE) as a comparison metric. Regardless of the task or model, we used the `scikit-learn` implementation for all metrics. More details about this stage of the pipeline can be found in APPENDIX B: HIRID-ICU-PIPELINE DETAILS.

## 5 Experiments

### 5.1 Settings

For all models, we tuned specific hyper-parameters using random search. Each randomly picked set of parameters was run with 3 different random initializations. We then selected hyper-parameters on the validation set performance for either AUPRC, B-Accuracy, or MAE. All models were trained with early stopping on the validation loss. Further details about hyper-parameters can be found in APPENDIX B: HIRID-ICU-PIPELINE DETAILS.

Because of the class imbalance existing in classification tasks, we considered balanced loss weights for all methods. However as further discussed in subsection 5.5, this technique was relevant only for the Patient Phenotyping task. For regression tasks, we min-max scaled the labels at training time to avoid exploding gradients.

### 5.2 Benchmarked Methods

In our proposed benchmark, we considered two groups of machine learning algorithms. The first group consists of regular machine learning algorithms, which as shown are highly effective for ICU-related tasks [40, 16, 20]. It is composed of a Gradient Boosting method with LightGBM [26] and Logistic Regression. The second group is focused on deep learning methods. We select the most commonly used sequence models for this group: Recurrent neural networks (LSTM [17] and GRU [8]), convolutional neural networks (CNN), in particular, temporal convolutional networks (TCN) [3] and Transformer models [43].

### 5.3 Benchmarking Models on High-resolution ICU Data

In this section, we compare the previously described methods on all tasks. While DL approaches are provided with the entire history for all time points, ML methods use only the values of the current step as an input. Thus one would expect the latter models to perform significantly worse due to the lower amount of information provided.

**Stay-Level Tasks**   When comparing methods on tasks requiring a single prediction after 24h (Table 3), we observe the superiority of LGBM with hand-extracted features. Transformers outperformed other DL methods but we observe a significant performance gap with the best ML method in B-Accuracy for Patient Phenotyping and AUPRC for ICU Mortality. Concerning GRU and LSTM, their performance is similar to TCN's for ICU Mortality. However, on the Patient Phenotyping task, they do not manage to outperform even logistic regression.

---

[8]https://scikit-learn.org/stable/modules/generated/sklearn.metrics.roc_auc_score.html
[9]https://scikit-learn.org/stable/modules/generated/sklearn.metrics.average_precision_score.html

Table 3: *Benchmark of methods for stay level tasks.*(Top rows) ML methods; (Bottom rows) DL methods. All scores are averaged over 10 runs with different random seeds such that the reported score is of the form $mean \pm std$. In bold are the methods within one standard deviation of the best one. Classification metrics were scaled to 100 for readability purposes.

| Task | ICU Mortality | | Patient Phenotyping |
|------|------|------|------|
| Metric | AUPRC (↑) | AUROC (↑) | B-Accuracy (↑) |
| LR | $58.1 \pm 0.0$ | $89.0 \pm 0.0$ | $39.1 \pm 0.0$ |
| LGBM | $54.6 \pm 0.8$ | $88.8 \pm 0.2$ | $40.4 \pm 0.8$ |
| LGBM w. Feat. | $\mathbf{62.6} \pm 0.0$ | $90.5 \pm 0.0$ | $\mathbf{45.8} \pm 2.0$ |
| GRU | $60.3 \pm 1.6$ | $90.0 \pm 0.4$ | $39.2 \pm 2.1$ |
| LSTM | $60.0 \pm 0.9$ | $90.3 \pm 0.2$ | $39.5 \pm 1.2$ |
| TCN | $60.2 \pm 1.1$ | $89.7 \pm 0.4$ | $41.6 \pm 2.3$ |
| Transformer | $61.0 \pm 0.8$ | $\mathbf{90.8} \pm 0.2$ | $42.7 \pm 1.4$ |

Table 4: *Benchmark of methods for online monitoring tasks.* (Top rows) ML methods; (Bottom rows) DL methods. All scores are averaged over 10 runs with different random seeds such that the reported score is of the form $mean \pm std$. In bold are the methods within one standard deviation of the best one. Classification metrics were scaled to 100 for readability purposes. MAE is in units ml/kg/h for Kidney Function and in hours for Remaining LOS.

| Task | Circulatory failure | | Respiratory failure | | Kidney func. | Remaining LOS |
|------|------|------|------|------|------|------|
| Metric | AUPRC (↑) | AUROC (↑) | AUPRC (↑) | AUROC (↑) | MAE (↓) | MAE (↓) |
| LR | $30.5 \pm 0.0$ | $87.6 \pm 0.0$ | $53.0 \pm 0.0$ | $65.4 \pm 0.0$ | N.A | N.A |
| LGBM | $\mathbf{38.9} \pm 0.3$ | $\mathbf{91.2} \pm 0.1$ | $58.5 \pm 0.1$ | $69.3 \pm 0.2$ | $\mathbf{0.45} \pm 0.00$ | $56.9 \pm 0.4$ |
| LGBM w. Feat. | $\mathbf{38.8} \pm 0.2$ | $\mathbf{91.2} \pm 0.1$ | $\mathbf{60.4} \pm 0.2$ | $\mathbf{70.8} \pm 0.1$ | $\mathbf{0.45} \pm 0.00$ | $57.0 \pm 0.3$ |
| GRU | $36.8 \pm 0.5$ | $90.7 \pm 0.2$ | $59.2 \pm 0.3$ | $70.1 \pm 0.2$ | $0.49 \pm 0.02$ | $60.6 \pm 0.9$ |
| LSTM | $32.6 \pm 0.8$ | $89.9 \pm 0.1$ | $56.9 \pm 0.3$ | $68.2 \pm 0.3$ | $0.50 \pm 0.01$ | $60.7 \pm 1.6$ |
| TCN | $35.8 \pm 0.6$ | $90.5 \pm 0.1$ | $58.9 \pm 0.3$ | $70.0 \pm 0.2$ | $0.50 \pm 0.01$ | $59.8 \pm 2.8$ |
| Transformer | $35.2 \pm 0.6$ | $90.6 \pm 0.2$ | $59.4 \pm 0.3$ | $70.1 \pm 0.2$ | $0.48 \pm 0.02$ | $59.5 \pm 2.8$ |

**Online Failure Predictions** For the continuous classification tasks, where the maximum sequence length extends from 288 steps to 2016, DL methods do not leverage the additional history information. Indeed, as shown in Table 4, for both Circulatory and Respiratory Failure, LGBM trained only on the current variables outperforms all DL methods. Among these methods, LSTM is the most impacted, as it has noticeably lower scores. Finally, for all continuous tasks, including regression discussed below, the improvement brought by hand-extracted features is not as significant. It suggests that statistical features, when extracted from the entire history, are less informative.

**Online Regression Tasks** The final set of tasks we benchmark are regression tasks (Table 4). As for the classification case, LGBM-based methods outperform DL methods, which, among them, have similar performance. In addition, we do not observe any improvement brought by our selection of hand-extracted features. Moreover, the overall performances of the proposed methods are relatively low. While a MAE of $0.45$ ml/kg/h for Kidney Function is only twice smaller than the median urine output rate, a 57h error in Remaining LOS is more than twice the median length-of-stay. We believe these low scores are due to the nature of the labels' distributions, which are both heavy-tailed as shown in APPENDIX A: DATASET DETAILS.

## 5.4  Behaviour of Deep Learning Approaches for Long Time-Series

One notable difference between the MIMIC-III benchmark [16] and our work is the data resolution. The resolution of our data being twelve-time higher leads to 2016 steps (1 week) sequences for online tasks. Thus, we explore if the increase of sequence length explains the decrease in performance of DL methods for continuous tasks.

**History Length**   One way to verify if DL methods leverage long-term dependencies in their prediction is to check if a decrease in the considered history impacts performance. We can achieve this for Transformers and TCN architectures, by respectively using local attention or fixing the number of dilated convolutions. In the results (Figure 2), we observe that both models do not use the additional information provided by early steps for the Circulatory failure task. It is in line with LGBM's lack of performance improvement when provided with history features on this task. For the Respiratory Failure task, where history features improve LGBM performance, shortening considered history impacts significantly both methods. TCN performance consistently decreases as history diminishes, whereas the Transformer model AUPRC first improves, almost closing the gap with LGBM, before also lowering. Thus, both DL models leverage history in the Respiratory Failure task. However, this also highlights known limitations of Transformers for long sequences [7] when the history exceeds 12h.

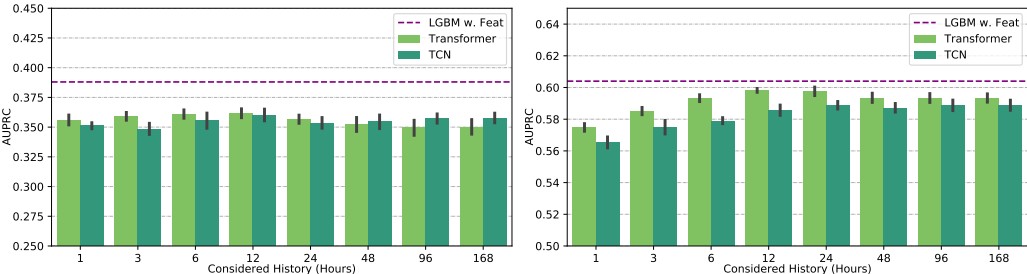

Figure 2: *Impact of history length on online classification performance.* (Left) Comparison in AUPRC for the Circulatory Failure task ; (Right) Comparison in AUPRC for the Respiratory Failure task. Error bars represent the standard deviation over 5 runs with different random initializations.

**Data Resolution**   Another approach to decrease the length of sequences is to reduce the data resolution. We compare all DL methods with a 1h prediction interval to assess the impact of data resolution on performance. This way, we can gradually lower the data resolution from 5min to 1h while preserving the same prediction time-steps. We report the result of this experiment in Figure 3. We observe that while TCN and Transformer performance are almost identical, GRU and LSTM are both impacted in opposite ways. GRU is noticeably better than LSTM on both tasks with a 5min grid, but as resolution lowers to 1h, this gap is significantly reduced.

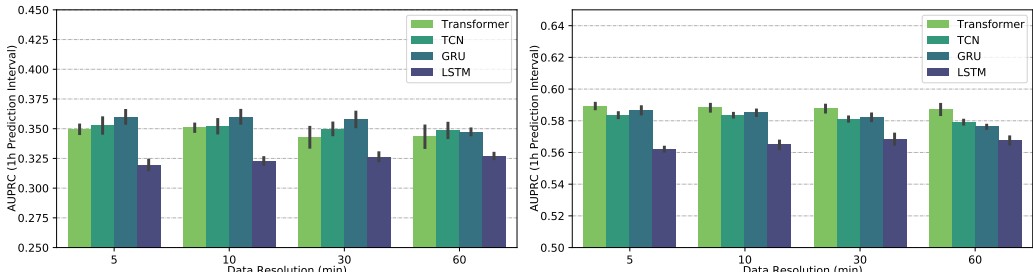

Figure 3: *Impact of data resolution on online classification performance.* (Left) Comparison in AUPRC for the Circulatory Failure task ; (Right) Comparison in AUPRC for the Respiratory Failure task. Error bars represent the standard deviation over 5 runs with different random initializations.

## 5.5   On Weighting Cross-entropy by Class Prevalence

All the tasks we define show a certain degree of imbalance and the class imbalance problem (CIP) is known to be highly challenging [25]. The most common approach to this problem is the use of class weights in the loss objective. For this ablation, we adopt the original idea from [27] by defining weights inversely proportional to each class prevalence. As shown in Table 5, for the multi-class task, it yields a significant improvement on the balanced accuracy. However, such a technique harms all binary classification tasks performances. It is particularly true for the highly imbalanced tasks, Circulatory Failure, and ICU Mortality.

Table 5: *Deltas in metrics of using balanced cross-entropy loss.* (Blue) Improvements over using no weights; (Red) Deterioration over using no weights.

| Task | ICU Mortality | | Phenotyping | Circulatory Failure | | Respiratory Failure | |
|---|---|---|---|---|---|---|---|
| Δ Metric | AUPRC (↑) | AUROC (↑) | B-Accuracy (↑) | AUPRC (↑) | AUROC (↑) | AUPRC (↑) | AUROC (↑) |
| LGBM w. Feat. | -1.3 | 0.0 | +4.3 | -4.2 | -2.6 | 0.1 | 0.0 |
| Transformer | -2.6 | 0.0 | +4.0 | -0.9 | 0.0 | -0.1 | -0.1 |

# 6  Discussion

In this paper, we provided an in-depth benchmark on the HiRID dataset and evaluated the behaviour of various machine learning models on diverse clinically relevant tasks developed in collaboration with intensive care clinicians. Our primary contribution is a full and reproducible preprocessing and machine learning pipeline and benchmark tasks on a public intensive care dataset, a necessary prerequisite for reproducible and comparable research in the future. We further evaluate current state-of-the-art machine and deep learning algorithms on these tasks establishing a baseline to compare future methods against. We consider this our second major contribution.

This work confirms previous results [20], that conventional machine learning models (i.e. boosted ensembles of decision trees) outperform current deep learning approaches on medical time series problems. Based on the experimental results we found that deep learning models do not lead to the same breakthrough performance increases as in other domains (such as NLP [10] or Computer Vision [11]). We believe the sparsity of the data and the imbalance of labels in both regression and classification tasks play an important role in this. For classification tasks, building a specific objective for highly imbalanced tasks such as Focal loss [32] might be a potential direction of research. For regression, a recent work has shown some promising leads for heavy-tailed regression tasks [44]. Moreover, HiRID introduces a novel high-resolution aspect in ICU data, that needs to be correctly taken into account. Thus, as for other sequence data, one possible explanation could be that when trained with extremely long sequences, models can not use the extracted features in the most effective way [46]. In the case of Transformers, to force the model to learn and extract useful patterns, various kinds of improvements could be made [41]. In particular, *learnable patterns* could be incorporated [38].

Our work goes beyond previous ICU time-series benchmarks (e.g. [16]) by using a more diverse set of tasks and a data set with a higher time resolution. As discussed earlier the set of clinical prediction tasks is diverse regarding the assessed organ systems, prevalence, and task type. An important limitation of our study is that HiRID is currently not the most frequently used and known ICU data set.

This work facilitates the future development of machine learning methods and standardized comparison of their performance on a diverse set of predefined tasks. It could contribute to solving today's problem of machine learning on medical time series not being comparable due to each work's unique datasets, preprocessing, and tasks definition. We hypothesize that methods developed and successfully evaluated on these tasks can also be successfully transferred to other specific medical time series problems.

This work also fills the gap between proposed machine learning approaches and their applications to ICU tasks. As a concrete example, COVID-19 is a big challenge for ICU patient monitoring. Important issues in this context are the uncertainty of the patient's prognosis as well as the prediction of the disease progression. COVID-19 is known to cause respiratory failure [31], one of the tasks studied in our benchmark, which is also the main cause for ICU admission and death [30, 39, 37, 18]. During the current COVID-19 pandemic, first attempts to construct a Respiratory Failure prediction model were already done such as [5], however, their data is available only for a limited audience, limiting reproducibility.

# 7   Conclusion

In this paper, we proposed an in-depth benchmark on time series collected from an Intensive Care Unit (ICU). In collaboration with clinicians, we defined several tasks relevant for healthcare covering different critical aspects of ICU patient monitoring. We provide a reproducible end-to-end pipeline to derive both data and labels, and a training setup to evaluate the final performance. We hope that this benchmark facilitates the construction and evaluation of machine learning methods for ICU data, and encourages reproducible research in this field.

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
