# OpenReview forum: "HiRID-ICU-Benchmark --- A Comprehensive Machine Learning Benchmark on High-resolution ICU Data"
_NeurIPS.cc/2021/Track/Datasets_and_Benchmarks/Round1 — NeurIPS 2021 Datasets and Benchmarks Track (Round 1)_

### Official Review · Reviewer_wf47 · 2021-06-28
**This work sets up six benchmark tasks based on existing ICU time series data and it needs improvements on details to be more user-friendly.**

**Rating:** 7
**Confidence:** 4
**Correctness:** Yes.
**Clarity:** The paper is well written and easy to…

**Strengths:**

The benchmark tasks in this work have several desired features:
- the tasks are highly clinical oriented and diverse (two regression tasks; three binary classification tasks and one multi-class task)

- the standard pre-processing pipeline is provided, which is critical for reproducibility

- open-sourced baselines

**Weaknesses:**

Major
1. There is a high requirement to access the benchmark dataset (e.g., need to join a course and take a few hours to train). Could you directly make the pre-processed data publicly available by Google Drive or Dropbox? If it is not allowed by the data holder, could you provide some simulated data to allow users to run and test your code?
2. It would be better to provide a more user-friendly tutorial on the GitHub repo. with a small subset of the (simulated) data, e.g., Jupyter-notebook.
3. Please provide the download link to baseline results (e.g., trained models, predictions).
4. Please provide specific explanations or direct links to the background of the dataset, which can release the burden for the newcomers that want to enter this domain. For example, what is the meaning of each variable? What are the clinical backgrounds/values of the benchmark tasks? How to understand the evaluation metrics?... Although these answers can be obtained by google or the dataset paper, it will be more convenient for future users if you provide them on the GitHub readme.

Minor
1. Please put Table 1 on Page 3 or on the top of Sec. 3.2.
Furthermore, what are the relationships of the six tasks? Please provide more motivations.
2. line 122: We provide more details about it in the README section of the Software Repository...
Do you mean the GitHub repo? https://github.com/ratschlab/HIRID-ICU-Benchmark/blob/master/README.md?  I do not find the preprocessing "details" in the README except for the command. Please add a footnote to specify the link of details.
3. Figure 1 Title: The explanation should follow the pipeline (i.e., Red, Blue, Green, Grey)
4. Add statistical significance analysis in Table 3 and Table 4.
5. Provided some data samples in the manuscript will be better.

**Additional Feedback:**

Please see the detailed comments in the weaknesses part.

**Documentation:**

Yes.

**Ethics:**

This work is based on public datasets. Thus, there are no ethical concerns.

**Relation To Prior Work:**

Yes. It was presented in Sec. 2.

**Summary And Contributions:**

This work provides a modular and standard pipeline to pre-process the public ICU time-series dataset (HiRID), creates six clinically relevant prediction benchmark tasks, and evaluates regular machine learning methods (LightGBM, Logistic Regression) and deep learning methods (LSTM, GRU, TCN, and Transformer).

---

> ### Author Response · Authors · 2021-07-14
> **Response to Reviewer wf47**
>
> Thank you for this in-depth review of our work. Your feedback was very helpful to improve the accessibility of our work. We detail our answers to the points raised below
>
> **Provide some simulated data to allow users to run and test the pipeline.**
> We are working on providing simulated data in the GitHub repo on which one can run the pipeline. It will be finished very soon, please check our repo for updates.
>
> **Provide a more user-friendly tutorial on the GitHub repo.**
> In addition to the simulated data, we already improved our *README* section in the GitHub repo. We added key resources for newcomers to the fields as well as more detailed instructions to run the code.
>
> **Provide the download link to baseline results (e.g., trained models, predictions)**
>  We provided all weights for all the models of the main experiment. As we ran each model for 10 random seeds we picked the weights with median performance. We also improved the package with direct function and associated scripts to evaluate these pretrained models.  Finally, we updated the *README* accordingly.
>
> **Provide specific explanations or direct links to the background of the dataset.**
> We added footnotes and references about this to *Appendix A: HiRID Dataset Details*. as well as a better reference to this appendix in the main paper. We also added a section named "Key Resources" to the *README* in the  GitHub repo.
>
> **Please put Table 1 on Page 3 or on the top of Sec. 3.2.**
> As for the footnotes 4 and 5 linked to this table, due to its size, we aren’t capable of moving it next to *Sec 3.2* on the same page. Though, we still changed the layout hoping it’s now clearer.
>
> **What are the relationships of the six tasks? Please provide more motivation.**
> The tasks are motivated and carefully chosen from a medical and machine learning viewpoint.
> From the medical point of view, our aim was to cover as many major organ systems as possible and also to have tasks regarding the overall patient state.
> We define tasks for the cardiovascular, kidney, and respiratory systems. These are three of the most important organ systems. For each organ system, we provide a task related to the main organ function. For other and missing organ systems e.g. neurological function, the liver function we do not have the necessary data available to define an appropriate endpoint. Mortality, Length of stay, and patient phenotyping are suitable tasks to capture the overall state of a patient.
> From the machine learning side, we aimed for diverse tasks as stated.
> We added some more details on the medical side to the paper in subsection 3.2.
>
> **line 122: We provide more details about it in the README section of the Software Repository... Do you mean the GitHub repo?**
> We refactored this sentence and added a footnote with the Github repo by reference first the *Appendix B: HiRID-ICU Pipeline Details*. We also improved the README accordingly.
>
> **Figure 1 Title: The explanation should follow the pipeline (i.e., Red, Blue, Green, Grey)**
> Thank you for the suggestion, we have changed the title of Figure 1 accordingly.
>
> **Add statistical significance analysis in Table 3 and Table 4.**
> As there are no overlaps between the best deep learning models and LGBM + Features, we believe that further statistical analysis beyond multiple runs isn’t required.
>
> **Provided some data samples in the manuscript will be better.**
> Due to the high number of existing features (231), building a compact humanly-understandable figure of a sample is not possible in our opinion. However, we agree being able to visualize the data is a key feature to better understand it. Thus, we are currently working on a visualization tool taking the form of a notebook.

---

> > ### Comment · Reviewer_wf47 · 2021-07-15
> > **Response to authors**
> >
> > Thanks for providing more details in the response. I accepted the authors' rebuttal and decided to increase the final rating. This dataset will provide more information for the medical and machine learning community.

---

### Official Review · Reviewer_X6z1 · 2021-07-02
**Solid contribution with need for more information before acceptance**

**Rating:** 8
**Confidence:** 3

**Strengths:**

**Significance:** this contribution is significant as it addresses an important need in the ICU space, namely a way to readily compare performance of different approaches with a standardized set of tasks and pre-processing. The temporal resolution may also enable improved performance over other datasets.

**Relevance:** the contribution is highly relevant to the broader community as managing ICU patients will only continue to be an important clinical need

**Accessibility and Accountability:** the documentation is mostly clear.

**Ethical/Social Implications:** managing ICU is an important social problem.

**Weaknesses:**

There are important details about the demographics of the participants and some of the methods that may limit the contribution. A major weakness of the paper is lack of clarity and details in the methods. In particular:

- it would be very helpful if the information about the different signals used could be included in the manuscript. I had to search for it in the documentation.  At least linking to here would be helpful: https://hirid.intensivecare.ai/Data-details-1ff9c433b9894904b1dbd7652be4b11c
- the authors should include more information about the pre-processing stage, and how they were able to verify that the pre-processing results were appropriate. This is especially important considering the varied nature of the signals. For example, I was wondering whether the heart rate data were extracted from e.g. continuous ECG measurements by the researchers in generation of the HiRID dataset. If this is beyond the scope of present work please include a line referring to the original HiRID paper.
- the authors should explain what they mean by "density of the measurement" (the rest of the features seem self-explanatory)
- the authors should discuss the scaling factors used for the continuous variables. How was the scaling conducted to insure comparison across participants without losing important information such as whether the patient is in tachycardia?
- there does not appear to be any summary of the demographics of the patients included in the dataset. Given that this dataset is intended to be used for machine learning, understanding how is included in the dataset and who is not is essential.


**Additional Feedback:**

The footnotes 4 and 5 appear to be on the incorrect page.

**Clarity:**

What is written in the paper is mostly clear, other than what was missing as described in weaknesses.

**Correctness:**

Claims made in the submission appear mostly correct. I focused on evaluation of the details of the ICU datasets and not as much on the model testing as that is not my background. The tasks seem diverse and appropriate.

**Documentation:**

The documentation appears clear, however there is important information missing as described in the weaknesses section. More information could be added about:

- intended uses
- hosting, licensing, and maintenance

**Ethics:**

The authors should include additional information about dataset demographics as described in weaknesses.

**Relation To Prior Work:**

Yes, the authors discuss previous datasets and attempts for tasks and benchmarks for ICU data in detail.

**Summary And Contributions:**

Here the authors present a number of contributions related to the HiRID-ICU database, including:

- a standardized pipeline for pre-processing, with two stages: one with just simple pre-processing (e.g. artifact removal) and one with pre-selected features for machine learning
- a series of tasks that can be used for comparison of different approaches to analyzing the ICU data (such as organ failure, mortality, etc.)
- benchmark results on the listed tasks above

---

> ### Author Response · Authors · 2021-07-14
> **Response to Reviewer X6z1**
>
> Thank you for this extensive review that has been very helpful to improve our work. Our response and implementation of the changes you required are provided below.
>
> **Provide information about different signals (variables) in the manuscript.**
> We added the part about this to *Appendix A: HiRID Dataset Details* with the URL for the official HiRID documentation (https://hirid.intensivecare.ai/) and with the URL to the table containing all information relative to each variable on the GitHub repo:   (https://github.com/ratschlab/HIRID-ICU-Benchmark/blob/master/preprocessing/resources/varref.tsv).
>
> **Provide more information about the pre-processing stage**
> The pre-processing steps are described in detail in the Data Preprocessing section in *Appendix A: HiRID Dataset Details*. We did not include the preprocessing stage information in the main paper due to length constraints. We made reference to the Appendix.
>
> **Provide the explanation of the meaning of "density of the measurement".**
> We added the additional explanation in subsection *4.2 Hand-engineered Feature Extraction* in the main paper.
>
> **Discuss the scaling factors used for the continuous variables.**
> We provided a further explanation about the scaling choice in subsection *4.1 Data Pre-processing*. The scaling was done globally using statistics from the training set ensuring that patient-specific attributes such as tachycardia can be recovered after scaling.
>
> **Add a summary of the demographics of the patients included in the dataset**
> We have provided a table with key demographic features in *Appendix A: HiRID Dataset Details*. We added a reference to it in the main manuscript (line 64).
>
> **Add more information about intended uses, hosting, licensing, and maintenance**
> We completed the README introduction with intended usage. We also added a reference to the license. Concerning hosting and maintenance we updated *Appendix B: HiRID-ICU Pipeline Details*.
>
> **The footnotes 4 and 5 appear to be on the incorrect page**
> Unfortunately, we weren’t able to move these footnotes as they belong to Table 1 which is too large to be on the same page.

---

> > ### Comment · Reviewer_X6z1 · 2021-07-19
> > **Response to authors**
> >
> > Thanks to the authors for their updates. These help clarify the contributions and have mostly addressed my concerns. My only minor comment is that race/ethnicity would be helpful to include in the demographic information if it is available, especially given important concerns regarding the use of primarily Caucasian participants in the development of ML models.

---

### Official Review · Reviewer_PyjH · 2021-07-02
**Timeseries ICU Dataset**

**Rating:** 6
**Confidence:** 4
**Correctness:** Yes
**Clarity:** Yes paper is will written.

**Strengths:**

One of the major problems with time series datasets is that unlike image datasets they are not standardized.  The results reported in the paper highly depend on the preprocessing step and generally machine learning researchers do not want to spend time processing data;  which leads to focus on images and language where there are many standard datasets for specific tasks eventually  ​leading to less research in this area.


This  paper  puts  the data in  machine  learning-friendly  format allowing them to focus on the research  part  making it extremely beneficial and will eventually lead to progress in research in this area.

The time series tasks proposed reflect real world challenges; also there is a variety of regression and classification tasks.

**Weaknesses:**

There is nothing particularly novel proposed by this paper. The dataset already exists, metrics are commonly for the mentioned tasks. Model comparison is pretty straight forward.



**Additional Feedback:**

Additional experiments comparing forecasting models might be interesting for example:
Forecasting accuracy at  different horizons.
Forecasting when training data length << test data length.




**Documentation:**

The github repository is easy to follow.
There is sufficient detail to support reproducibility.

**Relation To Prior Work:**

The difference between HiRID-ICU-Benchmark and MIMIC-III is compared.

**Summary And Contributions:**

The paper provides a  end-to-end pipeline for time-series analysis fo the HiRID dataset an ICU dataset.
This pipeline involves the preprocessing of data, proposing possible tasks where this dataset can be used, finally evaluating models.
Tasks proposed were a variety of classification and regression tasks.
The pipeline allows hand selection of feature.
The paper used the proposed pipeline  to compare LightGBM, Logistic Regression, LSTM, GRU,TCN and Transformer models.

---

> ### Author Response · Authors · 2021-07-14
> **Response to Reviewer PyjH**
>
> Thank you for taking the time to provide an in-depth review and useful comments.  We detail below our response to your main concerns.
>
> As you rightfully mentioned, raw HiRID data, metrics, and models employed in our work all already exist. However, we do not believe these constitute contributions to our work. Indeed, we first provide a set of diverse and clinically relevant tasks allowing us to evaluate multiple aspects of machine learning approaches. In addition, as pointed out in our “Related Work” section, while multiple open-source ICU datasets exist, common pre-processings and task definitions don’t, leading to unfair comparisons of models. We believe that with our unified pipeline we fill that gap by providing a fair benchmark to compare future works on.
>
> Concerning the additional ablations on the forecast horizon, we agree this is an interesting area to explore. We are currently working on it and will include them in the camera-ready version of the paper.

---

### Author Response · Authors · 2021-07-14
**Global Response to Reviewers**

We thank the reviewers for their time and effort in reviewing our paper and providing feedback. The consensus among the reviewers is that our work is well written (R PyjH, R X6z1, R wf47), ‘extremely beneficial’ (R PyjH) to the community by addressing “ an important need” (R X6z1) for “a standardized set of tasks and pre-processing”(R X6z1). Our work ensures crucial reproducibility through a “standard pre-processing pipeline”, “open-source baseline” (R wf47), and an “easy to follow” (R PyjH) documentation. According to all reviewers, the tasks we designed are “highly relevant” (R X6z1) to the field as they “reflect real-world challenges'' (R PyjH), are “highly clinical oriented”(R wf47), and diverse (R PyjH, R wf47).

There seem to be 3 main concerns which we address below.

**Improve documentation about different signals (variables) and demographics in the manuscript (meaning of each variable, clinical backgrounds/values of the benchmark tasks, etc).**
- We added a part about this to Appendix A: HiRID Dataset Details with the URL for the official HiRID documentation (https://hirid.intensivecare.ai/) and with the URL to the table containing all information relative to each variable on the GitHub repo:   (https://github.com/ratschlab/HIRID-ICU-Benchmark/blob/master/preprocessing/resources/varref.tsv). We also completed the GitHub README with the necessary resources concerning the dataset and metrics documentation.

- Summary statistics about the patient cohort are displayed in Appendix Table 1 (in Appendix A: HiRID Dataset Details). We now provide a reference to this table in the main manuscript.

**Provide a more user-friendly interface in the GitHub repo.**
- We improved the README section with further details on running the different parts of the pipeline.

- We added a section in the README containing useful links and information with respect to the data, models, and metrics used.

- To ease the understanding of the pipeline, we are working on providing simulated data. We will also add a subsection on the GitHub repo about simulated data generation and running the pipeline on it. It will be finished very soon, please check our repo for updates.

**Provide the download link to baseline results (e.g., trained models, predictions).**
- We added pretrained weights used for the experiments of the paper (median run whose performance is written in the result tables) to the repo.

- We improved the package by providing ready-to-use scripts to directly run and evaluate the pretrained models’ performances.
- We documented the usage in the new version of the README.

We also uploaded an updated version of the manuscript and supplementary materials that addressed these and other questions/notes.

---

### Decision · Program_Chairs · 2021-07-26

**Decision:**

Accept

**Comment:**

The reviewers all liked the paper.  The dataset will provide more information for the medical and machine learning community. The authors' response clarified some important points.  The authors are strongly invited to integrate these points in the final version.